# Molecular Interactions of Tannic Acid with Proteins Associated with SARS-CoV-2 Infectivity

**DOI:** 10.3390/ijms23052643

**Published:** 2022-02-27

**Authors:** Mohamed Haddad, Roger Gaudreault, Gabriel Sasseville, Phuong Trang Nguyen, Hannah Wiebe, Theo Van De Ven, Steve Bourgault, Normand Mousseau, Charles Ramassamy

**Affiliations:** 1Centre Armand-Frappier Santé Biotechnologie, 531 Boulevard des Prairies, Laval, QC H7V 1B7, Canada; mohamed.haddad@inrs.ca; 2Institute on Nutrition and Functional Foods, Laval University, Quebec City, QC G1V 0A6, Canada; 3Succursale Centre-Ville, Départment de Physique, Université de Montréal, Case Postale 6128, Montréal, QC H3C 3J7, Canada; roger.gaudreault@umontreal.ca (R.G.); gabriel.sasseville@umontreal.ca (G.S.); normand.mousseau@umontreal.ca (N.M.); 4Département de Chimie, Université du Québec à Montréal, 2101 Rue Jeanne-Mance, Montréal, QC H2X 2J6, Canada; nguyen.vo_thanh_phuong@uqam.ca (P.T.N.); bourgault.steve@uqam.ca (S.B.); 5Département de Chimie, Université McGill, 3420 Rue University, Montréal, QC H3A 2A7, Canada; hannah.wiebe@mail.mcgill.ca (H.W.); theo.vandeven@mcgill.ca (T.V.D.V.)

**Keywords:** SARS-CoV-2, COVID-19, molecular dynamics, polyphenols, RBD, TMPRSS2, 3CLpro

## Abstract

The overall impact of severe acute respiratory syndrome coronavirus 2 (SARS-CoV-2) on our society is unprecedented. The identification of small natural ligands that could prevent the entry and/or replication of the coronavirus remains a pertinent approach to fight the coronavirus disease (COVID-19) pandemic. Previously, we showed that the phenolic compounds corilagin and 1,3,6-tri-O-galloyl-β-D-glucose (TGG) inhibit the interaction between the SARS-CoV-2 spike protein receptor binding domain (RBD) and angiotensin-converting enzyme 2 (ACE2), the SARS-CoV-2 target receptor on the cell membrane of the host organism. Building on these promising results, we now assess the effects of these phenolic ligands on two other crucial targets involved in SARS-CoV-2 cell entry and replication, respectively: transmembrane protease serine 2 (TMPRSS2) and 3-chymotrypsin like protease (3CLpro) inhibitors. Since corilagin, TGG, and tannic acid (TA) share many physicochemical and structural properties, we investigate the binding of TA to these targets. In this work, a combination of experimental methods (biochemical inhibition assays, surface plasmon resonance, and quartz crystal microbalance with dissipation monitoring) confirms the potential role of TA in the prevention of SARS-CoV-2 infectivity through the inhibition of extracellular RBD/ACE2 interactions and TMPRSS2 and 3CLpro activity. Moreover, molecular docking prediction followed by dynamic simulation and molecular mechanics Poisson–Boltzmann surface area (MMPBSA) free energy calculation also shows that TA binds to RBD, TMPRSS2, and 3CLpro with higher affinities than TGG and corilagin. Overall, these results suggest that naturally occurring TA is a promising candidate to prevent and inhibit the infectivity of SARS-CoV-2.

## 1. Introduction

Severe acute respiratory syndrome coronavirus 2 (SARS-CoV-2), a zoonotic coronavirus first identified in China, has led to the worldwide coronavirus disease (COVID-19) pandemic with more than 400 million known cases of infection and 5.8 million deaths as of February 2022. Mutated SARS-CoV-2 variants are emerging with increased infectivity, facilitating their spread [1]; e.g., the SARS-CoV-2 B.1.617.2 (Delta) variant possesses a higher replication rate and transmissibility than B.1.1.7 (Alpha) [2]. The B.1.1.529 (Omicron) variant with 30 mutations has also emerged [3]. Though vaccination campaigns have been implemented in many countries, there is still an urgency to develop effective and accessible therapeutics.

SARS-CoV-2-induced infection involves multiple steps, from extracellular to transmembrane and finally intracellular. In addition to the spike protein receptor angiotensin-converting enzyme 2 (ACE2), transmembrane protease serine 2 (TMPRSS2) and the viral 3-chymotrypsin like protease (3CLpro) (also known as the main protease (M^pro^)) are required for cell entry and replication, respectively. SARS-CoV-2 infection is controlled by the opening of the spike protein receptor binding domain (RBD), where a conformational transition from a ‘down’ to an exposed ‘up’ state, gated by an N-glycan shield at position N343, occurs to bind with ACE2 [4]. TMPRSS2 located on the membrane of the host cell cleaves and activates the SARS-CoV-2 spike protein, leading to structural rearrangements, membrane fusion, and the release of the viral RNA into the cytoplasm of the host cell [5,6,7]. Once the virus has taken control of the host cell, it uses its viral replication and transcription complex (RTC) as well as the cell’s own translation machinery for its replication. Then, newly replicated virions exit the cell via exocytosis and spread to neighboring cells and throughout the body.

Many strategies to mitigate the mechanisms of action of SARS-CoV-2 have been investigated [8,9,10,11,12,13,14,15,16,17], including the usage of small molecules to inhibit viral entry and replication [18,19]. Natural products or their derivatives make up 49.2% of the 1881 new drugs developed from January 1981 to September 2019 [10,20], and numerous researchers have turned to naturally occurring polyphenols, as their therapeutic potential has been documented for antiviral uses [21] and as antiviral-drug candidates for SARS-CoV-1 and SARS-CoV-2 [22,23,24,25].

The protective effects of polyphenols against SARS-CoV-2, especially through the inhibition of binding between the spike protein and the ACE2 receptor, have been substantially investigated [26]. For example, Yang et al. [27] showed that corilagin dose-dependently blocks SARS-CoV-2 RBD binding and suppresses the infectious property of RBD pseudo-typed lentivirus in HEK293 cells overexpressing hACE2. Our recent numerical and experimental results showed that two natural polyphenols, corilagin and 1,3,6-tri-O-galloyl-β-D-glucose (TGG), can disrupt the extracellular interactions between ACE2 and SARS-CoV-2 spike wild-type as well as mutated RBD proteins [28]. Experimentally, both molecules bound preferably to the spike protein, whereas only very weak binding was observed with ACE2. Hence, the physiological side effects induced by ACE2 inhibition would likely remain very limited [28].

Since the early COVID-19 pandemic, most studies have focused on the inhibition of RBD/ACE2 binding [29], but natural polyphenols could also target two proteolytic enzymes—the transmembrane TMPRSS2 and the intracellular 3CLpro. The monomer of SARS-CoV-2 3CLpro has three structural domains: I (residues 10–96), II (residues 102–180), and III (200–303) [30]. Tahir ul Qamar et al. [31] screened a library containing 32,297 potential antiviral phytochemicals and traditional medicinal compounds and showed that one of the binding areas of 3CLpro is located on the active sites of the catalytic dyad residues His41 and Cys145. This dyad located at the interface of His41 in Domain I and Cys145 in Domain II [32], separated by 3.6 Å, is the optimum distance to initiate H-bonding [30]. Using molecular dynamics (MD) and in vitro methods, Loschwitz et al. [33] showed that corilagin could inhibit 88% of 3CLpro activity. Du et al. [34] showed (−)-epicatechin-3-O-gallate (ECG) to be a potent inhibitor of 3CLpro with an IC_50_ of 0.847 ± 0.005 μM. Wang et al. [24] have identified tannic acid (TA) as a potent inhibitor of both TMPRSS2 and 3CLpro. TA can inhibit 100% of 3CLpro activity with an IC_50_ value ranging from 2.1 μM [35,36] to 9 μM [37]. An inhibition of 77 ± 1% of 3CLpro activity could be achieved with a mixture of 5 μM TA combined with 20 μM puerarin, 20 μM daidzein, and 20 μM myricetin [37].

TA (C_76_H_52_O_46_) is a naturally occurring polyphenolic compound found in several plants with similar properties to the two phenolic ligands previously studied, TGG (C_27_H_24_O_18_) and corilagin (C_27_H_22_O_18_) [28,38,39,40], the latter possessing promising medicinal properties [41] and very low toxicity in mice even at high dosages [42]. For instance, corilagin has been described as a non-steroidal anti-inflammatory and an antioxidant [43] with antihypertensive properties [44]. It has two joined phenolic rings (R3-R6) which make it more rigid than TGG. On the other hand, the less studied TGG is closely related to two tetra-O-galloyl-β-D-glucose molecules that were identified as promising therapeutic compounds against SARS-CoV-1 [22,45]. Turning to TA, this molecule exhibits very low toxicity in mice with a LD_50_ of 3500 to 5000 mg/kg body weight and is flexible and hydrolysable, whereas its potential metabolites may also show inhibitory effects against SARS-CoV-2.

In this study, we investigate experimentally the interactions between natural polyphenolic ligands—particularly TA, TGG, and corilagin—with proteins involved in the relevant steps for cellular entry and replication of the virus—RBD (N501Y) (the most frequent variant at the start of the study), TMPRSS2, and 3CLpro. We use a combination of experimental methods (biochemical enzyme-linked immunosorbent assay (ELISA), enzymatic assay, surface plasmon resonance (SPR), and quartz crystal microbalance with dissipation monitoring (QCMD)) and numerical tools (molecular docking, molecular dynamics, and molecular mechanics Poisson–Boltzmann surface area (MMPBSA) free energy calculations). Based on the biochemical data, we then concentrate on TA and its biophysical and numerical interactions with RBD (N501Y), TMPRSS2, and 3CLpro. Overall, our work highlights the potential role of TA in protection against SARS-CoV-2 infection by inhibiting the extracellular RBD/ACE2 interactions, the activities of the transmembrane TMPRSS2, and intracellular 3CLpro enzymes.

## 2. Results

### 2.1. Inhibitory Effect of Polyphenols on the Binding between RBD (N501Y) Spike Protein and Human ACE2

We first investigate the ability of seven, bioactive, natural compounds to inhibit the interaction between RBD (N501Y) spike protein and human ACE2 using ELISA binding assays. These compounds, namely TA, TGG, corilagin, and four other compounds: pelargonidin-3-O-glucoside, malvidin-3-O-glucoside, cyanidin-3-O-glucoside, and peonidin, have been found to be effective against SARS-CoV-2 binding to ACE2 in silico [24,32,46,47]. Among the seven tested compounds, only TA, TGG, and corilagin show a significant inhibitory effect, up to 50% at 10 µM (Figure 1A). Interestingly, TA reduces the binding between RBD (N501Y) spike protein and ACE2 by up to 95%.

A dose-dependent effect from 0.1 to 5 μM of TA, corilagin, and TGG, which are the focus of the rest of this work, shows a significant inhibition from 0.1 µM (Figure 1B–D).

#### Biophysical Characterization of TA/RBD Interactions

Considering the efficacy of TA, we then focus our biophysical characterization on TA. We perform SPR measurements to quantify the binding kinetics of TA to RBD (Figure 2A). The recombinant protein RBD (N501Y) is covalently immobilized on a carboxymethylated dextran sensor chip (CM5) using an amine coupling strategy. Representative sensograms show that TA, with a concentration ranging between 0.1 and 80 µM, binds avidly to the immobilized protein. The SPR curves of the TA/RBD complex, i.e., association (300 s) and dissociation (1200 s) are well defined, and the complex is very stable with a dissociation rate constant of 1.07 × 10^−4^ s^−1^. Fitting the sensograms to a one-site (1:1 molecular ratio) binding model leads to an equilibrium dissociation constant (K_D_) in the nanomolar range, i.e., 41.98 nM for the formation of the TA/RBD complex (Appendix A).

We also perform QCMD experiments to observe the adsorption of TA onto RBD (Figure 2B). First, RBD is adsorbed to the gold surface of the QCMD sensor, with an average total mass of 5.0 ± 2.4 mg/m^2^ of RBD (N501Y), and then TA solutions (10 to 500 µM) are flowed over the protein-coated sensor for 30 min, with results showing that the amount of TA adsorbed increases with TA concentration. At this time, equilibrium adsorption is not reached, thus the adsorption isotherm is only approximate. Moreover, the kinetics of TA adsorption on RBD are shown in Appendix A. Consequently, both SPR and QCMD methods show similar trends—TA binds to RBD with a high affinity.

### 2.2. Polyphenols and TMPRSS2 Enzyme

#### 2.2.1. Effect of TA, TGG, and Corilagin on TMPRSS2 Activity

To determine the capacity of TA, TGG, and corilagin to inhibit the activity of TMPRSS2, we incubate the enzyme with different concentrations of these polyphenols from 0.1 to 100 µM. Our results show that TA from concentrations of 10 µM and higher has a significant and dose-dependent inhibitory effect on TMPRSS2 activity (Figure 3A). Interestingly, at 50 µM, TA reduces TMPRSS2 activity by up to 50%. However, no inhibition effect is observed with TGG and corilagin in this range of concentrations (Figure 3B,C).

#### 2.2.2. Biophysical Characterization of TA/TMPRSS2 Interactions

We concentrate on the biophysical characterization of TA due to its considerable impact. The recombinant TMPRSS2 is covalently immobilized on a CM5 sensor chip, and the kinetics of the binding of TA to TMPRSS2 are analyzed by SPR (Figure 4A). Fitting the sensograms to a one-site (1:1 molecular ratio) binding model leads to a K_D_ of 11.68 nM, characterized with a tight binding, i.e., low dissociation. As observed for RBD, the binding of TA to the immobilized TMPRSS2 is very tight, with a dissociation rate constant of 1.51 × 10^−4^ s^−1^ (Appendix A).

QCMD is used to measure the adsorption of TA onto TMPRSS2 (Figure 4B). TMPRSS2 is adsorbed to the gold QCMD sensor, with an average total mass of 6.3 ± 2.6 mg/m^2^ of TMPRSS2. TA solutions (10 to 500 µM) are then flowed over the TMPRSS2 protein-coated surface for 30 min, and TA adsorption is observed to increase at higher concentrations of TA. The kinetics of TA adsorption on TMPRSS2 as a function of time are also shown in Appendix A.

### 2.3. Polyphenols and 3CLpro Enzyme

#### 2.3.1. Effect of TA, TGG, and Corilagin on 3CLpro Activity

To determine the capacity of TA, TGG, and corilagin to inhibit the activity of 3CLpro, we incubate the enzyme with different concentrations of these polyphenols from 0.1 to 100 µM. We find that TA has the most potent inhibition, with a significant inhibitory effect starting from 0.1 µM. Interestingly, at 10 µM, TA reduces the 3CLpro activity by up to 95% (Figure 5A). We also observe a significant inhibitory effect of TGG and corilagin on 3CLpro activity from 50 µM (Figure 5B,C).

#### 2.3.2. Biophysical Characterization of TA/3CLpro Interactions

Given the intensity of its inhibitory effect, the binding of TA to 3CLpro is analyzed by SPR with a CM5 sensor chip coated with the protein (Figure 6A). For the kinetics of the formation of the TA/3CLpro complex, fitting the sensograms to a one-site (1:1 molecular ratio) binding model leads to a K_D_ in the nanomolar range, i.e., 57.47 nM. As observed for RBD and TMPRSS2, the binding of TA to the immobilized 3CLpro is very tight, with a dissociation rate constant of 1.76 × 10^−4^ s^−1^ (Appendix A).

We also perform QCMD experiments to observe the adsorption of TA onto 3CLpro (Figure 6B). First, 3CLpro is adsorbed to the gold surface of the QCMD sensor, with an average total mass of 4.2 ± 2.9 mg/m^2^ of 3CLpro, and then TA solutions (10 to 500 µM) are flowed over the protein-coated sensor for 30 min, wherein TA adsorption is observed to increase with higher TA concentrations. In addition, Appendix A shows the kinetics of TA adsorption on 3CLpro as a function of time. Overall, both SPR and QCMD methods show similar trends, i.e., TA binds to 3CLpro with a high affinity.

### 2.4. Molecular Modelling

#### 2.4.1. Molecular Docking and Dynamics of TA/RBD Complex

To further elucidate the molecular basis of our experimental observations, we turn to molecular docking to identify the most stable conformations. Docking of TA to RBD with AutoDock VINA leads to docked positions with a highest binding affinity of −6.8 kcal/mol (pose 1), followed by −6.7 (pose 2), −6.7 (pose 3), and −6.6 (pose 4) kcal/mol (Appendix A). For example, Figure 7C shows pose 1 for the TA/RBD (N501Y) complex with the TA ligand localized away from the N501Y mutation.

Molecular docking with AutoDock VINA considers only limited protein and ligand conformational dynamics. Yet, molecular flexibility is critical for reliable and predictable characterization [33,48]. Thus, we perform 1000-ns MD simulations starting from the four best predicted binding poses of TA/RBD(N501Y) as generated by VINA in order to allow local rearrangements of both the protein and ligand. As calculated over the last 250 ns, the MMPBSA binding free energy for the various TA/RBD (N501Y) complexes varies from −70 to −41 kcal/mol, with the most negative number indicating a more stable and preferable binding (Table 1).

Figure 8A shows that no major conformational structural changes take place with respect to both TA binding for pose 1 and the overall protein structure between the initial and final configurations of the 1000-ns MD run. Moreover, Appendix A shows that in poses 1 and 4, with the lowest binding free energy, TA is located in the β-sheets region of the receptor-binding motif (RBM), whereas in poses 2 and 3 TA is located in the RBM loop; in all poses TA is far from the N501Y mutation (Figure 7C).

The LigPlot interaction map between TA and RBD for pose 1 (Figure 9A) shows that TA is forming five hydrogen bonds (H-bonds) with RBD (Phe490, Ser494, Gly496, Val503, and Tyr505), and that the complex is also stabilized by 16 other contacts, including eight amino acids with hydrophobic side chains (Tyr351, Ala352, Tyr449, Leu452, Leu455, Tyr473, Tyr489, and Leu492) (Figure 9B and Appendix A). The interaction maps from the four best poses are shown in Appendix A. For the TA/RBD complexes, pose 3, stabilized by three H-bonds and six other contacts including four amino acids (aa) with hydrophobic side chains, is the least stable at −41 kcal/mol. On the other hand, pose 4 shows the highest binding value (−70 kcal/mol) with 10 H-bonds and 12 other contacts, including five aa with hydrophobic side chains.

Except for pose 3, where the MD run shows TA moving from the loop to the beta-sheets, all poses remain stable over 1000 ns, with the ligand strongly associated with the RBM region. This suggests that TA could also alter the binding of RBD (N501Y) with ACE2, consistent with our previous work in which TGG and corilagin ligands were investigated [28].

#### 2.4.2. Molecular Docking and Dynamics of TA/TMPRSS2 Complex

Averaged over the 750 to 1000 ns time interval, the MD MMPBSA free energy of TA with TMPRSS2 varies from −71 (pose 4) to −33 (pose 2) kcal/mol (Table 1), with pose 1 shown in Figure 10A. The LigPlot interaction map between TA and TMPRSS2 shows that TA in pose 1 forms seven H-bonds with TMPRSS2 (Ser84, Lys85, Asp90, Glu134, Lys135, Arg158, and Lys212), and that the complex is also stabilized by 12 other contacts, including five amino acids with hydrophobic side chains (Ala40, Tyr159, Val160, Leu164, and Trp206) (Figure 10B and Appendix A). The interaction maps for the four best poses are shown in Appendix A. Similar trends are observed for all TA/TMPRSS2 complexes. With three H-bonds and further stabilized by eight other contacts, including four aa with hydrophobic side chains, pose 2 shows a binding free energy of −33 kcal/mol, while pose 4 is the most attached, with a binding free energy of −71 kcal/mol associated with five H-bonds and 16 other contacts, including five aa with hydrophobic side chains. All these structures are stable, as validated by the evolution of the root-mean-square deviation (RMSD) of TA/TMPRSS2 over the 1000 ns interval and the root-mean-square fluctuation (RMSF) averaged over the last 250 ns of the MD run (Figure 8B, which superposes the initial and final configurations for pose 1, and Appendix A, which shows RMSD and RMSF for all poses).

#### 2.4.3. Molecular Docking and Dynamics of TA/3CLpro Complex

The MMPBSA binding free energy of TA with 3CLpro is −65 kcal/mol for the most stable pose generated, with TA in the 3CLpro pocket (Table 1, Figure 11A). Over the 750 to 1000 ns interval, TA forms four H-bonds with 3CLpro (Cys145, Cys22, Ala191, and Gln192) (Appendix A). These H-bonds are computed using PyMOL with the same cut-off distance as used for previous molecules. The LigPlot map shows that the complex is also stabilized by 17 other contacts, including five amino acids with hydrophobic side chains (Val42, Met49, Leu50, Met165, and Leu167) (Figure 11B, Appendix A). The H-bond length of the TA/3CLpro complex is approximately 3Å, similar to that of the TA/RBD and TA/TMPRSS2 complexes. Moreover, the RMSD of TA/3CLpro is stable at 0.3 nm within 100 to 300 ns, then moves to 0.35 to 0.40 nm within the 300 to 1000 ns trajectory (Appendix A). This is mostly due to the movement of flexible loops away from the ligand as shown in Figure 8C, which overlaps the initial (t = 0) and final (t = 1000 ns) configurations.

In summary, the enzymatic activity assays show that TA outperforms the inhibitory effect of all other studied phenolic ligands on the association of RBD with ACE2, followed by TGG and corilagin (Figure 1). The inhibitory effects of TA and TGG are shown to be concentration dependent. SPR experiments show that the studied TA/protein interactions are dose-dependent, supporting the binding of TA with RBD (N501Y), TMPRSS2, and 3CLpro, with their dissociation constants in the low nanomolar range (between 11.68 and 57.47 nM). In addition, QCMD experiments suggest a high affinity adsorption of TA to the three proteins. Regarding the MD followed by MMPBSA calculations, the lowest binding free energies of the TA/RBD, TA/TMPRSS2, and TA/3CLpro complexes are −70, −71, and −65 kcal/mol, respectively. Interestingly, the percentage of stabilizing contacts within the TA/protein complexes, e.g., amino acid hydrophobic side chains, lies between 38% to 67% for TA/RBD, 31% to 50% for TA/TMPRSS2, and 29% for TA/3CLpro, suggesting that hydrophobic interactions play a significant role in TA/protein complexes. Overall, although the calculated free energy depends on the size of the ligand, preventing a direct comparison with smaller molecules, our results suggest that the binding between TA and RBD, TMPRSS2, and 3CLpro is very strong.

## 3. Discussion

The COVID-19 pandemic caused by SARS-CoV-2 infection remains a global public health concern. Despite several available vaccines, infection is still largely uncontrolled in the presence of certain variants, including the SARS-CoV-2 B.1.1.7 lineage with the N501Y mutation [52]. In the search for effective COVID-19 treatments, many drug candidates have failed. To date, remdesivir has been approved by the U.S. Food and Drug Administration (FDA), but it is not recommended by the World Health Organization due to insufficient evidence of its effectiveness against COVID-19. The mutagenic ribonucleoside molnupiravir [53] and the 3CLpro inhibitor nirmatrelvir/PF-07321332 (Paxlovid^®^) [54] are two recently FDA-approved oral antivirals for the treatment of COVID-19. However, there are some safety concerns regarding molnupiravir, as it can cause mutations in host cells [55]. Furthermore, nirmatrelvir/PF-07321332 is highly dependent on CYP3A for clearance and could interfere with strong inducers of CYP3A4 or with immunosuppressive agents [56] and should be used in combination with ritonavir, an inhibitor of CYP3A; however, as CYP3A represents the most important cytochrome P450 for drug metabolism in critical tissues such as the gastrointestinal tract and liver, its inhibition could lead to hepatoxicity. Neither molnupiravir nor nirmatrelvir/PF-07321332 are authorized for use as pre-exposure or post-exposure prophylaxis for the prevention of COVID-19; moreover, the projected cost is around $500 (Paxlovid^®^) to $700 (molnupiravir) per person for a 5-day course. As a result, it is critical to identify a multitargeted, low-cost drug that has few or no side effects for the treatment of SARS-CoV-2-induced infection. To this end, harnessing natural products could be a promising strategy to identify effective compounds to fight COVID-19.

The entry of the virus into host cells is a critical event for cellular infection and replication cycles. The spike (S) protein of SARS-CoV-2 facilitates viral entry into target cells. To achieve this, the S1 subunit of the spike RBD engages ACE2 as the entry receptor and employs the cellular serine protease TMPRSS2 for S-protein priming, which entails S-protein cleavage at the S1/S2 boundary and allows the fusion of the viral and cellular membranes, a process driven by the S2 subunit [5,57,58]. During the replication of the virus, the SARS-CoV-2 3CLpro enzyme is required for proteolytic maturation of the viral polyproteins in order to form the RNA replicase–transcriptase complex, which is critical for both viral transcription and replication processes [59]. 3CLpro is thus considered as the main target for virus inhibition. Therefore, inhibition of RBD/ACE2 interaction and TMPRSS2 and 3CLpro activities represent relevant strategies to inhibit the cellular entry of the virus, its replication, and therefore its infectivity.

Since the environment and the type of surface influence the attachment of the virus and its persistence on human cell surfaces [8], we used a multidisciplinary approach in this study, i.e., experimental methods (experimental binding, enzymatic assays, SPR, QCMD) combined with numerical tools (protein-ligand docking, molecular dynamics, MMPBSA calculations) to demonstrate the beneficial effects of a potential preventive drug. The selected methods cover many aspects of infection and the COVID-19 disease: enzymatic inhibition (Figure 1, Figure 3 and Figure 5), binding kinetics of association/dissociation rate constants (Figure 2, Figure 4 and Figure 6), adsorption of TA over immobilized recombinant proteins (Appendix A), molecular structures (Appendix A), flexibility and/or rigidity of natural phenolic ligand/protein complexes, types of interactions (e.g., the number of H-bonds and solvent-accessible surface area (SASA)) (Appendix A), binding residues (Appendix A), docking binding affinity (Appendix A), and MMPBSA free energy calculations (Appendix A).

While we consider three ligands—TGG, corilagin, and TA—our main focus is on the third for a number of reasons. The biosynthetic pathway of TA starts from D-glucose, then, after 10 galloylation reactions (gallic acid esterified to a single glucose moiety), yields decagalloyl-glucose. TA is highly soluble in water and possesses 25 phenolic hydroxyl groups which can initiate H-bonds, 10 hydrophobic phenolic rings essential for π-π stacking with aromatic amino acid residues, and one beta-D-glucose sugar ring which shows some flexibility between chair and boat/skew-boat conformations (Appendix A) [38,39]. The 10 galloyls also increase conformational flexibility, which plays a significant role in ligand binding. In addition, the size and higher molecular weight of TA (1701.18 g/mol) compared to the other ligands suggest a higher contact surface area with a targeted protein.

Our results demonstrate that TA, TGG, and corilagin can inhibit the binding between the spike protein RBD (N501Y) and the human ACE2 receptor as well as reduce the activities of the enzymes TMPRSS2 and 3CLpro, with TA being the most potent ligand, able to reduce the binding by up to 95% and enzyme activities by 60% to 70%. Compared to our results, Wang et al., [24] found that TA led to a higher inhibition of TMPRSS2 activity (IC_50_ 2.31 μM versus 50 μM in this study) but a lower inhibition for 3CLpro (IC_50_ 13.4 μM versus 1μM in this study). This discrepancy is likely based on the choice of methodology. Wang et al. used a fluorescence resonance energy transfer (FRET)-based enzyme activity assay, while we used an ELISA-like assay. Interestingly, in both studies, TA was identified as the most potent inhibitor among the tested phenolic compounds. Our results also confirm that TA is the most promising SARS-CoV-2 3CLpro inhibitor, as it was previously found among a library of 720 natural products, with an IC_50_ of 3 µM compared to 3-isotheaflavin-3-gallate (IC_50_ of 7 μM) and theaflavin-3, 3′-digallate (IC_50_ of 9.5 μM) [36].

TA is a well-known protein precipitating agent [60] that can modify the reaction parameters and activities of proteins [61]. The concentration effect of TA observed here indicates that the results described in this study are not attributed to the protein precipitating effects of TA.

The efficiency of TA observed in the binding and enzymatic experiments is also confirmed by two additional and complementary methods, the SPR measurements (a gold standard in biomolecular interaction technology) and the QCMD experiments. Although both SPR and QCMD methods provide the kinetics of adsorption and interaction, SPR allows for the measurement of the association and dissociation rate constants, whereas QCMD gives the adsorbed weight of protein and ligand as a function of time. The main difference in the techniques is that the SPR information is obtained from an optical biosensor that monitors the change in refractive index of the surface interface that occurs during the binding process, whereas the QCMD information is obtained from a change in resonance frequency of a quartz crystal sensor. In principle, rate constants can be found from both methods (from mathematical models), but this is more difficult for QCMD because the exact adsorbed mass is unknown (the co-adsorbed water overestimates the dry mass of ligand and protein [62]). Our SPR results show dissociation constants (K_D_) in the low nanomolar range for TA with all studied proteins, e.g., 41.98 nM for TA/RBD (N501Y) (Figure 2A). From SPR measurements of the binding kinetics of TA to the recombinant protein TMPRSS2, we determined a K_D_ of 11.68 nM (Figure 4A). This is in sharp contrast to Wang et al. [24], who reported a K_D_ of 1.77 µM for TA/TMPRSS2 (1.56 to 25 µM TA), with an IC_50_ of 2.31 µM. Our SPR method also finds a K_D_ in the low nanomolar range for TA/3CLpro binding, 57.47 nM (Figure 6A), which is lower than previously described (0.78 to 25 µM TA) [24]. A recent review summarizes numerous results on SPR biosensing of SARS-CoV-2 [63,64]. In this work, both SPR and QCMD methods show high affinity between TA and RBD, TMPRSS2, and 3CLpro.

Building on molecular dynamics trajectories for these complexes, we calculate the binding free energy using the MMPBSA method. Our results show that the TA/protein binding free energies for various poses are: −41 to −70 kcal/mol for TA/RBD, −33 to −71 kcal/mol for TA/TMPRSS2, and −65 kcal/mol for TA/3CLpro (Table 1). However, Pitsillou et al. [32] found an MMPBSA binding free energy of cyanidin-3-O-glucoside/3CLpro of −50.8 and −42.1 kcal/mol with protomers A and B, respectively, whereas Singh et al. [65] found the highest free energy among three phenolic ligands (mangiferin, glucogallin, and phlorizin) with 3CLpro (−9.65 ± 3.33 kcal/mol), while the molecular weights (MW) of these ligands (332.26 to 449.38 g/mol) are four to five times smaller than that of TA (1701.18 g/mol). In addition, Patil et al. [66] reported the binding free energy of 3CLpro with rutin (−30.96 ± 5.56 kcal/mol), amentoflavone (−32.14 ± 4.73 kcal/mol), and remdesivir (−26.67 ± 3.39 kcal/mol), where their MW are 610.5, 538.5, and 602.6 g/mol, respectively. Albohy et al. [67] also reported a free energy of the 3CLpro/acaciin complex of −27.61 kcal/mol, where the MW of acaciin is 592.5 Da, i.e., about three times smaller than that of TA. Interestingly, there are discrepancies from literature, e.g., Gogoi et al. [68] reported an MMPBSA ECG/3CLpro binding free energy of −46.03 ± 6.48 kcal/mol, in agreement with Loschwitz et al. [33] (−41.5 ± 5.1 kcal/mol), though the MW of ECG is 442.4 g/mol. Overall, when compared to the aforementioned references, most of our results show a stronger binding for all TA/protein complexes, with free energies from −33 to −71 kcal/mol (Table 1) suggesting a higher affinity between TA and the three proteins assessed in this work than for the smaller molecules previously studied with the same MMPBSA method. This may be due in part to the higher molecular weight of TA; for example, most of the smaller phenolic molecules previously studied have a MW below 500 g/mol. As no quantitative expression exists to relate molecular weight and binding energy, we cannot normalize our results to ensure a better theoretical comparison at this time.

In addition to the targets described in this study, TA could also contribute to the management of SARS-CoV-2-induced symptoms through its ability to control oxidative stress [69] and inflammatory reactions [70]. Patients with COVID-19 presented higher total oxidative and reduced glutathione levels [71,72,73], leading to an increase in oxidative stress, which contributes to viral pathogenesis by stimulating inflammatory mechanisms through the activation of different pathways, particularly the nuclear factor kappa B (NF-κB) leading to the cytokine storm [74].

Not only does TA have effects on the main proteins involved in SARS-CoV-2 infection, but hydrolysable TA metabolites could also be involved in these processes, e.g., gallic acid or its gut microbial metabolite pyrogallol, which have been shown to interact with 3CLpro [75]. The pyrogallol group of certain polyphenols could serve as an electrophile to bind to Cys145 [76].

The present study provides useful insights for the use of natural-product-derived molecules for the development of drugs to prevent or to minimize SARS-CoV-2 infection. Among seven promising polyphenols tested, we found that TA was the most potent natural compound to prevent the RBD (N501Y) spike protein/human ACE2 interaction, as well as to inhibit the activities of TMPRSS2 and 3CLpro. As with most polyphenols, its efficacy is contingent on its bioavailability, which depends on multiple factors, including the properties of the molecule itself, intestinal microbiota, pH values, and consumption alongside other compounds. Furthermore, it is also important to take into account the inter-variability between individual COVID-19 cases. However, we anticipate that this study will pave the way for novel, tannin-based, small molecules to become more efficacious and selective anti-COVID-19 therapeutic compounds. The challenge now is to move forward and to translate these potential preclinical findings into effective therapeutic agents for the treatment of COVID-19 disease and its complications.

### Some Limitations of this Study

Although we used a combination of experimental methods and numerical tools, the limitations are the absence of the complex microenvironment with a pseudovirus or SARS-CoV-2 infection in physiological conditions. The SPR and QCMD methods apply certain assumptions to treat the experimental data—a Langmuir binding model was applied to the SPR data (Section 4.3); when treating the QCMD data, the Sauerbrey equation assumes a thin, rigid film, which may not hold true for large proteins, and QCMD also detects co-adsorbed water molecules, which was accounted for by an assumption that proteins and TA adsorbed with the same weight fraction of bound water (Section 4.4.2 and 4.4.3). As for the computational techniques, MMBPSA free energy calculations are limited by solvent approximation and sampling (Section 4.5.5). Despite these approximations in the experimental and computational methods, the isolation of the ligands and proteins in question serves to demonstrate the convergence of the effectivity of TA relative to other polyphenols as an agent against SARS-CoV-2 infectivity.

## 4. Materials and Methods

### 4.1. Products

TA (1,2,3,4,6-penta-O-{3,4-dihydroxy-5-((3,4,5-trihydroxybenzoyl)oxy)benzoyl}-D-glucopyranose), with molecular formula C_76_H_52_O_46_ and MW of 1701.18 g/mol, was obtained from Millipore Sigma (Burlington, MA, USA). The powder material, C.A.S. 1401-55-4, was natural in origin and ACS grade.

TGG (1,3,6-tri-O-galloyl-β-D-glucose), with molecular formula C_27_H_24_O_18_ and MW of 636.46 g/mol, was obtained from MuseChem (Fairfield, NJ, USA). The powder material, C.A.S. 18483-17-5, is natural in origin with purity 98.23%.

Corilagin (β-1-O-galloyl-3,6-(R)-hexahydroxydiphenoyl-D-glucose), with molecular formula C_27_H_22_O_18_ and MW of 634.45 g/mol, was obtained from Cayman Chemical (Ann Arbor, MI, USA). The powder material, C.A.S. 23094-69-1, was natural in origin and with purity >98%.

Pelargonidin-3-O-glucoside, malvidin-3-O-glucoside, cyanidin-3-O-glucoside, and peonidin were obtained from Extrasynthese (Z.I. Lyon Nord, France).

Bovine serum albumin (BSA), sulfuric acid (H_2_SO_4_), dimethyl sulfoxide (DMSO), N-hydroxysuccinimide (NHS), and ethanolamine-HCl were obtained from Sigma-Aldrich (Oakville, ON, Canada).

Phosphate buffered saline (PBS) was purchased from Wisent Bioproducts (Saint-Jean-Baptiste, QC, Canada).

The 2019-nCoV spike protein host cell receptor binding domain (RBD) (N501Y), with a MW of 26.5 kDa, was purchased from Creative Biomart (Shirley, NY, USA). The protein was expressed at the Arg319-Phe541 region in human embryonic kidney (HEK293) cells with a His-tag at the C-terminal and with purity >90% as determined by SDS-PAGE.

Biotinylated recombinant human ACE2 with purity >95% and camostat mesylate compound were both purchased from R&D Systems (Minneapolis, MN, USA).

Recombinant human TMPRSS2 protein (106–492 aa), with a MW of 44.8 kDa, was purchased from Creative Biomart. The protein was produced by a yeast expression system with a His-tag at the N-terminal and with purity >90% as determined by SDS-PAGE. The TMPRSS2 substrate BOC-Gln-Ala-Arg-AMC was purchased from Bachem (Bu-bendorf, Switzerland).

Severe acute respiratory syndrome coronavirus 2 3C-like protease (SARS-CoV-2 3CL Protease: 1–306 aa) (full length) with a MW of 34 kDa was purchased from BPS Bioscience (San Diego, CA, USA), and was expressed in an *E. coli* system, with purity >90%. GC367 compound was also purchased from BPS Bioscience.

3-(N,N-dimethylamino)-propyl-N-ethylcarbondiimide (EDC) was acquired from Thermo Fisher (Waltham, MA, USA).

### 4.2. Biochemical Assays

#### 4.2.1. SARS-CoV-2 RBD (N501Y) Spike Protein and Human ACE2 Binding Inhibitor Assay

The enzyme-linked immunosorbent assay (ELISA) established and detailed in our previous study [28] was used to determine the capacity of TA, TGG, corilagin, pelargonidin-3-O-glucoside, malvidin-3-O-glucoside, cyanidin-3-O-glucoside, and peonidin to inhibit binding between the RBD (N501Y) spike protein and human ACE2. Briefly, RBD (N501Y) at a concentration of 0.5 μg/mL was coated on ELISA plates and kept overnight at 4 °C. Plates were then rinsed, and wells were blocked with 1% BSA in PBS for 1 h at 37 °C. After the washing step, biotinylated human ACE2 protein at a concentration of 0.25 μg/mL was added to each well and incubated at 37 °C for 1 h, followed by the addition of diluted peroxidase-conjugated streptavidin to each well and further incubation at 37 °C for 30 min. Chromogenic substrate solution was added to each well and incubated at 37 °C for another 30 min. The enzymatic reaction was stopped by 50 µL of H_2_SO_4_ (2 N) solution, and the absorbance was then read at 450 nm using the Synergy HT multi-mode microplate reader (BioTek Instruments, Winooski, VT, USA). For the competition assay, polyphenols were incubated with immobilized RBD (N501Y) spike protein for 1 h at 37 °C before the addition of ACE2. To note, a concentration response curve for ACE2 (0.015 to 2 μg/mL) was established to confirm a concentration-dependent increase of the absorbance at 450 nm (Appendix A).

#### 4.2.2. TMPRSS2 Enzymatic Assay

The activity of the TMPRSS2 enzyme was determined according to the protocol described by Shrimp et al. [77] with some modifications. Briefly, the human recombinant TMPRSS2 enzyme was diluted to 10 ng/µL with the assay buffer. In a 96-well plate, 30 µL of the diluted enzyme was pre-incubated with 10 µL of different concentrations of polyphenols for 30 min. The enzymatic reaction was initiated by adding the TMPRSS2 substrate (BOC-Gln-Ala-Arg-AMC) at the final concentration of 10 µM. The fluorescence was monitored for 1 h at 5 min intervals with the excitation and the emission wavelengths fixed at 360 nm and 460 nm, respectively, using the Synergy HT multi-mode microplate reader. For control conditions, TMPRSS2 and its fluorescent substrate were incubated with either H_2_O, which was used to dilute TA, or DMSO, which was used to dilute TGG and corilagin. A 10 µM camostat mesylate compound, the inhibitor of TMPRSS2, was used for negative control conditions (Appendix A). Each fluorescence value was subtracted from the blank obtained from wells containing H_2_O or DMSO and 10 µM of the substrate.

#### 4.2.3. CLpro Enzymatic Assay

The enzymatic assay of the SARS-CoV-2-specific 3CLpro was carried out according to Mody et al. [78]. Briefly, 3CLpro–MBP tagged enzyme was diluted to 10 ng/µL with its assay buffer. In a 96-well plate, 30 µL of the diluted enzyme was pre-incubated with 10 µL of different concentrations of polyphenols for 60 min. The enzymatic reaction was initiated by adding the fluorescent substrate at the final concentration of 50 µM. After incubation for 18 h at room temperature, the fluorescence value obtained with excitation at 360 nm and emission at 460 nm was subtracted from the blank as described above. Negative controls were obtained with 100 µM of GC367 compound, a standard 3CLpro inhibitor (Appendix A).

#### 4.2.4. Statistical Analysis

Data were analyzed using the GraphPad Prism program. For the inhibitory effects of polyphenols on the interaction between SARS-CoV-2 spike protein RBD (N501Y) and human ACE2 and on the enzymatic activity of TMPRSS2 and 3CLpro, statistical analyses were performed using one-way ANOVA followed by the Tukey post hoc test. A p-value less than 0.05 was considered statistically significant.

### 4.3. Surface Plasmon Resonance (SPR)

SPR analyses were performed using a Biacore T200 instrument (GE Healthcare Bio-Sciences AB, Uppsala, Sweden). Recombinant RBD (N501Y) (319–541 aa), human TMPRSS2 (106–492 aa), and 3CLpro (1–306 aa) proteins (Appendix A) were each immobilized on a carboxymethylated dextran CM5 sensor chip (GE Healthcare) using an amine-coupling strategy. Briefly, the sensor chip surface was activated with a 1:1 mixture of N-hydroxysuccinimide (NHS) and 3-(N,N-dimethylamino)-propyl-N-ethylcarbondiimide (EDC). Recombinant protein solutions (20 μg/mL) were injected at a flow rate of 20 μL/min using a PBS + 0.05% Tween running buffer (150 mM NaCl, pH 7.4) to reach a level of immobilization of 200 RU. Surfaces (protein and reference) were blocked by the injection of an ethanolamine–HCl solution. The binding kinetics of TA over the immobilized recombinant protein sensor chip were evaluated in PBS + 0.05% Tween buffer with increasing polyphenol concentrations (1 to 80 µM) at a flow rate of 20 μL/min. Association time was set at 300 s and dissociation time was extended up to 1200 s. The sensor chip surface was regenerated by injecting 10 μL of 50 mM NaOH solution at a flow rate of 20 μL/min. Binding sensograms were obtained by subtracting the reference flow cell (without protein). Experiments were performed at least in duplicate and data analysis was performed using the BIA evaluation software package (v1.0, GE Healthcare) and fitted to a one-site (1:1 molecular ratio) Langmuir binding model.

### 4.4. Quartz Crystal Microbalance with Dissipation Monitoring (QCMD)

#### 4.4.1. QCMD Procedure

QCMD adsorption experiments were performed using a QSense Explorer (Biolin Scientific, Gothenburg, Sweden) with gold-coated quartz crystal sensors (QSX 301, Biolin Scientific). Prior to each experiment, sensors were cleaned by rinsing five times with DI water, then soaking in 2% Hellmanex^®^ III solution (Sigma-Aldrich) while in a sonication bath for 20 min. Sensors were then rinsed 10 times with DI water, once with ethanol, 10 times with DI water, then dried under a flow of air. Finally, sensors were placed in a UV/ozone chamber for 20 min.

To coat the sensor with protein, a 100 µL drop of 1 µg/mL protein solution (RBD (N501Y), TMPRSS2, or 3CLpro in PBS, pH 7.4) was pipetted carefully onto the working surface of the sensor. The protein was allowed to adsorb from the drop for 30 min, after which the remaining solution was removed by pipette and the sensor was dried under a gentle flow of air. The sensor was inserted into the QCMD flow module, and PBS was flowed over the surface using a peristaltic pump at a flowrate of ~100 µL/min until the resonance frequency of the crystal stabilized (|∆f| < 0.1 Hz/min).

TA adsorption to the protein-functionalized sensor was studied by flowing TA solutions (10 to 500 µM in PBS, pH 7.4) continuously over the sensor. All test solutions were prefiltered through a 0.45-µm syringe filter. The resonance frequency and dissipation shift of the crystal were measured for 30 min.

Desorption of mass from the sensor was evaluated by flowing PBS for 15 min following TA adsorption, but no appreciable desorption of TA was observed in all experiments. All experiments were performed at room temperature (22 °C) and at least in duplicate.

The relationship between the shift in resonance frequency of an oscillating quartz crystal and the mass deposited on its surface is given by the Sauerbrey equation [79]:∆m = − (C/n) ∆f(1)
where C is the sensitivity constant (17.7 ng cm^−2^ Hz^−1^ for a 5 MHz quartz crystal), n is the overtone number (1, 3, 5, 7, or 9), and ∆f is the shift in resonance frequency at the specified overtone n [80,81]. The third overtone (n = 3) was used in all experiments. To account for variability in the mass of protein adsorbed to the sensor, the quantity of TA adsorbed was normalized to the amount of protein adsorbed on the sensor. The ∆m was converted to a molar quantity of TA and divided by the amount of protein (in moles) adsorbed after drop coating, resulting in a dimensionless molar ratio.

#### 4.4.2. Water Content in the Protein Layer Adsorbed on the QCMD Sensor

When particles adsorb to a QCMD sensor, water molecules within (intrinsic) and between particles in the adsorbed layer are also sensed in the frequency shift [62,82]. Thus, the reported mass of protein adsorbed to the sensor, e.g., 5.0 ± 2.4 mg/m^2^ for RBD (N501Y), is an overestimate of the actual amount of protein (dry basis) adsorbed.

Nevertheless, monolayer coverage of the sensor surface by protein can be estimated geometrically by approximating the RBD (N501Y) protein as a sphere with a diameter of 3.65 nm (radius of gyration of approximately 1.825 nm calculated with GROMACS v2021.2, 23 February 2022, https://www.gromacs.org (Appendix A). Since a random deposition of spheres forming a monolayer corresponds to 55% coverage of the available surface area [83], a monolayer of RBD (N501Y) is estimated to be 1.78 mg/m^2^ according to the following calculations. Consequently, this suggests a weight fraction of approximately 35% for the protein and 65% water, although this fraction will be altered when TA adsorbs onto the protein layer, e.g., from protein conformational changes.

Mass of one RBD (N501Y) protein:(2)$$20,340\frac{g}{mol}\left(\frac{1\;mol}{6.022\times 10^{23}\;molec\;}\right)=3.377\times 10^{-20}\frac{g}{molec}$$

Area occupied by one RBD (N501Y) protein:(3)$$\frac{\pi {\left(\frac{3.65\;nm}{2}\right)}^{2}}{molec}=10.5\frac{nm^{2}}{molec}$$

Monolayer of RBD (N501Y) surface coverage:(4)$$0.55\left(\frac{3.377\times 10^{-20}\frac{g}{molec}}{10.5\frac{nm^{2}}{molec}}\right){\left(\frac{10^{9}\;nm}{m}\right)}^{2}\left(\frac{10^{3}\;mg}{g}\right)=1.78\frac{mg}{m^{2}}$$

#### 4.4.3. Molar Ratio of TA/Protein on the Sensor Surface

The amount of TA adsorbed onto the protein layer after 30 min is reported as a dimensionless molar ratio of TA to RBD, TMPRSS2, or 3CLpro (Figure 2B, Figure 4B and Figure 6B, respectively). The mass of the adsorbed protein layer varies between experiments (e.g., 5.0 ± 2.4 mg/m^2^ for RBD (N501Y)), as does the amount of TA adsorbed to the protein layer; this normalization accounts for the variance in adsorbed protein.

Moreover, it is likely that the amount of H_2_O molecules associated with TA (H_2_O:TA) differs from the amount of H_2_O molecules with the protein (H_2_O:protein) on a weight basis. For example, although it has been estimated that about 43 H_2_O molecules form a monolayer around gallic acid [84], a building block of TA (D-glucose after 10 galloylation reactions), only about five H_2_O molecules form stronger directional bonds with the hydrophilic part of GA, similar to what was reported experimentally by Martinez et al. [85]. However, here we hypothesize a similar water weight fraction between TA and protein to normalize and estimate the dimensionless molar (TA/protein) ratio. Under this assumption, a water weight fraction of 65% corresponds to approximately 175 H_2_O molecules per TA molecule, or ~17 H_2_O per galloyl group of TA. This number is higher than the reported five H_2_O molecules in close contact, but lower than the 43 associated H_2_O molecules in bulk water, thus it may be a reasonable approximation for the water fraction upon TA adsorption to the protein layer. Alternatively, there could be five H_2_O per galloyl group with the remaining water located between TA molecules.

### 4.5. Molecular Modeling

The structure of TA was calculated using the PM3 semiempirical method [86,87]. The most stable structures of the other two ligands, TGG and corilagin, were previously calculated using molecular modeling and PM3 semiempirical delocalized molecular orbital theory (DLMO) by one of the authors of this publication [38,39]. For this work, the TA.xyz, TGG.xyz, and corilagin.xyz structures obtained from the PM3 method were converted to be compatible with GROMACS (.pdb), using ACPYPE, 22 December 2021 (https://bmcresnotes.biomedcentral.com/articles/10.1186/1756-0500-5-367).

#### 4.5.1. Molecular Docking

Molecular docking between the proteins and TA was done using AutoDock VINA (v1.1.2, 27 December 2021, https://vina.scripps.edu) [88]. Protein flexibility was considered by performing the docking on the center of the clusters, representing a minimum of 5% of the sampled population, while ligand flexibility was taken into account by VINA’s methodology. The docking was constrained to specific regions of interest on the protein (active sites), e.g., residues 438 to 506 for RBD (N501Y). The protocol to generate the structure of RBD (N501Y), starting from the experimental crystal RBD/ACE2 complex (PDB:6M0J), was recently described [28]. For TMPRSS2, the region of interest is residue LYS87, equivalent to LYS392 from Huggins et al. [89]; consequently, our docking was targeted on this region using a box of 18 × 20 × 18 Å in the x, y, and z directions, respectively. With regard to 3CLpro (PDB:6LU7), the region of interest is the catalytic dyad residues H41 and C145 [33,90,91]; hence, our docking was constrained to the same region using a box of 18 × 18 × 18 Å in the x, y, and z directions, respectively. All three proteins were docked with the TA ligand. An exhaustiveness parameter of 100 was used, and a maximum energy difference of 40 kcal/mol between the best and the worst binding affinity was permitted. The PDB files were converted to PDBQT using AutoDock Tools (v1.5.6) [92,93] to be compatible with VINA.

#### 4.5.2. MD Simulations

To begin, a 500-ns MD simulation was performed on each protein (RBD (N501Y), TMPRSS2, and 3CLpro) separately, as well as on the ligand (TA). Each molecular system was prepared with the following steps: (1) the system underwent an energy minimization in vacuum using the steepest descent (SD) and conjugate gradient (CG) algorithms; (2) the system was then inserted into a periodic dodecahedron box which was then solvated with water molecules (TIP3P) (distance from the wall: 1 nm); (3) ions were added to the system to neutrality; (4) the system underwent a second energy minimization using the SD and CG algorithms while restraining all non-hydrogen atoms of the protein with harmonic restraints; (5) the system was equilibrated in the NVT ensemble at 300 K over 10 ns while maintaining the previous harmonic restraints; (6) the system was equilibrated in the NPT ensemble at 1 bar over 10 ns while maintaining the previous harmonic restraints; and (7) the system underwent a full molecular dynamics (MD) simulation without any harmonic restraints.

All simulations were run with GROMACS v2021.2 [94]. The AMBER14sb forcefield was used to determine the parameters of the molecules. The system was neutralized with the addition of counter ions (Na^+^ and Cl^−^). The Nosé–Hoover thermostat was used with a coupling constant (period of temperature fluctuations at equilibrium) of 0.1 ps to maintain the temperature constant at 300 K [95,96]. The Parrinello–Rahman barostat was used with a coupling constant (period of pressure fluctuations at equilibrium) of 2.0 ps to maintain the pressure constant at 0.987 atm [97]. These temperature and pressure values are in agreement with those used for AMBER14sb’s parameterization as well as the ones in our experiments. A cutoff of 1 nm was applied to both van der Waals and electrostatic interactions, the latter being computed using Particle Mesh–Ewald [98,99]. LINCS [100] and SETTLE [101] were used to constrain bond lengths and water geometry, respectively.

#### 4.5.3. Protein–Ligand Simulations

Following VINA molecular docking simulations, we used the highest negative binding affinity to launch MD simulations on each complex (composed of the ligand, TA, bound to a protein, RBD (N501Y) (333–526 residues), TMPRSS2 (1–234 residues), or 3CLpro (1–306 residues)) (Appendix A). A 1000-ns MD simulation was launched on each ligand/protein system according to the previously described protocol. In addition, the generalized AMBER forcefield (GAFF) [102] was used to determine the parameters of the ligand. The RESP protocol [103] from ANTECHAMBER [102,104] was used to determine the partial charges of the ligand.

#### 4.5.4. Analysis

The analysis of MD simulations was performed using the built-in GROMACS tools as well as in-house scripts. The RMSD and the RMSF were analyzed to determine conformational changes and to determine the convergence interval—the interval on which the rest of the analysis would be conducted. The SASA was computed for the residues located at the region of interest on each protein. Secondary structures were determined using DSSP [105]. Daura’s algorithm was used for clusterization [106]. The visualization of the molecules as well as the MD trajectory were done using PyMOL (v2.5.0, 23 February 2022, https://pymol.org) [107]. Then, LigPlot+ v2.2, 23 February 2022 (https://www.ebi.ac.uk/thornton-srv/software/LigPlus/) was used to visualize the protein/ligand interactions [50,51].

#### 4.5.5. Binding Free Energy

The MMPBSA method is a simple approach to estimate the binding free energy; it does, however, make a few crude approximations—the solvation is considered implicitly, thus possibly neglecting crucial water molecules at the binding site, and the entropic part of the equation is often neglected (as in this study) [108].

The MMPBSA protein/ligand binding free energy (∆G_bind_) is defined by:∆G_bind_ = < G_RL_ − (G_R_ + G_L_)_RL_ >(5)
where G_RL_, G_R_, and G_L_ are the free energies of the receptor/ligand complex, receptor, and ligand, respectively. We used a single trajectory MMPBSA computation—the conformations of the complex (RL), receptor (R), and ligand (L) were all taken from a unique MD trajectory. The bracket pair < > represents an ensemble average over all receptor/ligand conformations. More specifically, the free energy G is estimated according to:∆G = U + G_solvation_ − TS,(6)
where U is the internal energy, computed using the AMBER14sb forcefield field; G_solvation_ is the solvation free energy and is usually decomposed into a polar part, computed by solving the Poisson–Boltzmann equation, and a non-polar part that depends on the SASA; T is the temperature, and S is the entropy [108].

MMPBSA computations were done with the g\_mmpbsa utility [49], which uses APBS [109] to compute the polar part of the solvation free energy. The dielectric constants of the solute and solvent were set to 2 and 80, respectively. The surface tension (gamma) was set to 0.0226778 kJ/(mol/Å^2^), and the temperature was set at 300 K. The results were computed from the convergence interval of the ligand–protein MD simulations using 40 ps snapshots. A 500-step bootstrap analysis was used to compute the average and standard deviation of the free energy.

## 5. Conclusions

Natural products or their derivatives account for 49.2 percent of the 1881 new drugs developed in the last four decades. Small molecules have been extensively tested against SARS-CoV-2 since the outbreak began. Current and upcoming SARS-CoV-2 variants necessitate a continuous reevaluation of the efficacy of currently available treatments. In this study, combined experimental methods (ELISA, enzymatic assay, SPR, and QCMD) and computational methods (protein-ligand docking, molecular dynamics, and MMPBSA calculations) concluded that TA outperformed other polyphenols in terms of inhibition of SARS-CoV-2 infectivity by disrupting the virus’s extracellular RBD/ACE2 interactions, TMPRSS2 cellular entry, and intracellular 3CLpro replication. Overall, our findings show that the use of naturally occurring TA may be a useful strategy for preventing SARS-CoV-2 infectivity. As TA is a natural product obtained from plants, some of which are recognized for their use in food, it appears that its incorporation into therapeutic practice will be of great relevance. Notably, one of the benefits of considering TA as a therapeutic is that it has a good safety profile without the potential to cause certain major side effects. The upcoming phase of development should be to conduct laboratory model and clinical trials on COVID-19 patients to evaluate the possibility of reducing virus replication and clinical symptoms.

## Figures and Tables

**Figure 1 ijms-23-02643-f001:**
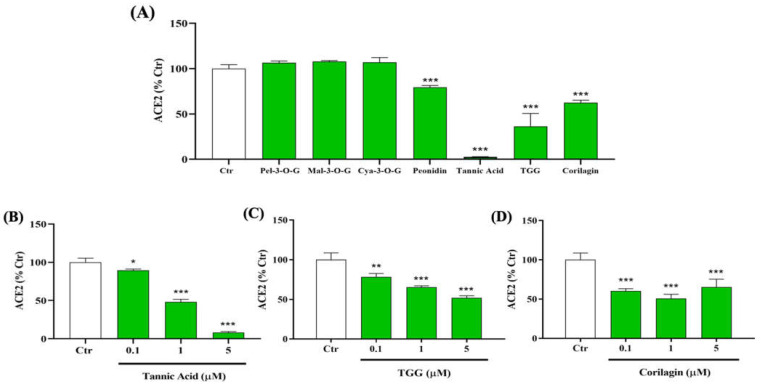
Inhibitory effects of different polyphenols on the interaction between SARS-CoV-2 spike protein receptor binding domain (RBD (N501Y)) and human angiotensin-converting enzyme 2 (ACE2). (**A**) 10 µM of pelargonidin-3-O-glucoside (Pel-3-O-G), malvidin-3-O-glucoside (Mal-3-O-G), cyanidin-3-O-glucoside (Cya-3-O-G), peonidin, tannic acid (TA), 1,3,6-tri-O-galloyl-β-D-glucose (TGG), and corilagin were tested to evaluate their ability to inhibit the binding of immobilized spike protein (0.5 µg/mL) to human, biotin-labeled ACE2 (0.25 µg/mL) by using an enzyme-linked immunosorbent assay (ELISA). Dose effect inhibition of 0.1, 1, and 5 µM (**B**) TA, (**C**) TGG, and (**D**) corilagin. The absorbance of ACE2 (0.25 µg/mL) at 450 nm was set to 100%. Results are expressed as mean ± SD (n = 3). Statistical analysis was performed using one-way ANOVA followed by the Tukey post hoc test with * *p* < 0.05, ** *p* < 0.01, *** *p* < 0.001 compared to ACE2 (0.25 µg/mL).

**Figure 2 ijms-23-02643-f002:**
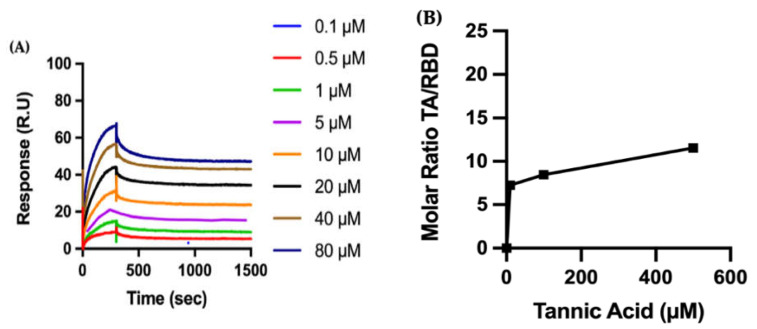
Biophysical characterization of the molecular interaction between TA and RBD: (**A**) Binding of polyphenol TA to immobilized RBD by surface plasmon resonance (SPR). The recombinant protein RBD (N501Y) is immobilized on a carboxymethylated dextran (CM5) sensor chip, and increasing concentrations of TA are injected to evaluate binding kinetics. (**B**) RBD is adsorbed to a gold quartz crystal microbalance with dissipation monitoring (QCMD) sensor, and various concentrations of TA are flowed over the surface for 30 min. TA adsorption is shown by the dimensionless molar ratio of adsorbed TA from solution to adsorbed RBD. The initial slope is a measure of the affinity of TA to RBD.

**Figure 3 ijms-23-02643-f003:**
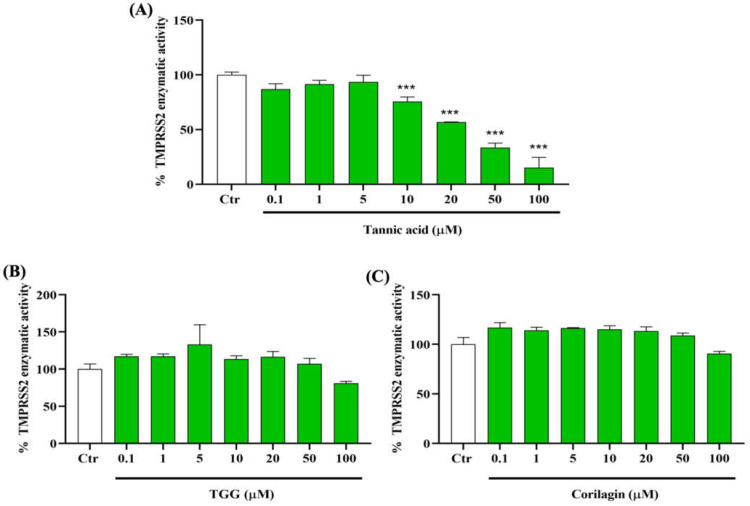
Inhibitory effects of TA, TGG, and corilagin on human transmembrane protease serine 2 (TMPRSS2) activity. The effects of different concentrations (0.1 to 100 µM) of (**A**) TA, (**B**) TGG, and (**C**) corilagin are tested on the activity of TMPRSS2. The fluorescence units in control conditions are considered as 100%. Blank values are subtracted from all the readings before the conversion into percentage of activity. Results are expressed as mean ± SD (n = 3). Statistical analysis is performed using one-way ANOVA followed by Tukey post hoc test with *** *p* < 0.001 compared to positive control wells.

**Figure 4 ijms-23-02643-f004:**
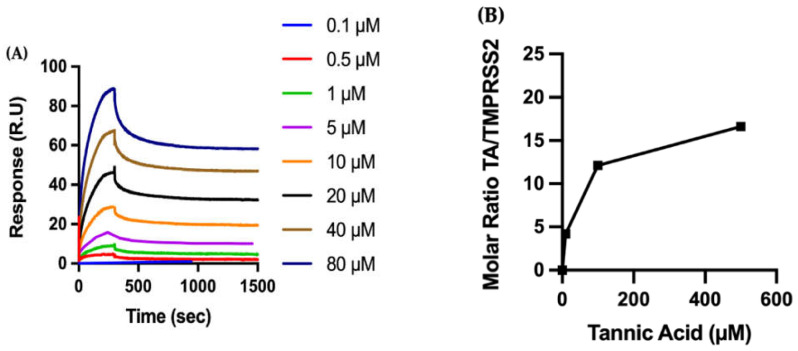
Biophysical characterization of the molecular interactions between TA and TMPRSS2. (**A**) The recombinant protein TMPRSS2 is immobilized on a CM5 sensor chip, and increasing concentrations of TA are injected to evaluate binding kinetics by SPR. (**B**) TMPRSS2 is adsorbed to a gold QCMD sensor, and various concentrations of TA are flowed over the surface for 30 min. TA adsorption is expressed by the dimensionless molar ratio of adsorbed TA to adsorbed TMPRSS2.

**Figure 5 ijms-23-02643-f005:**
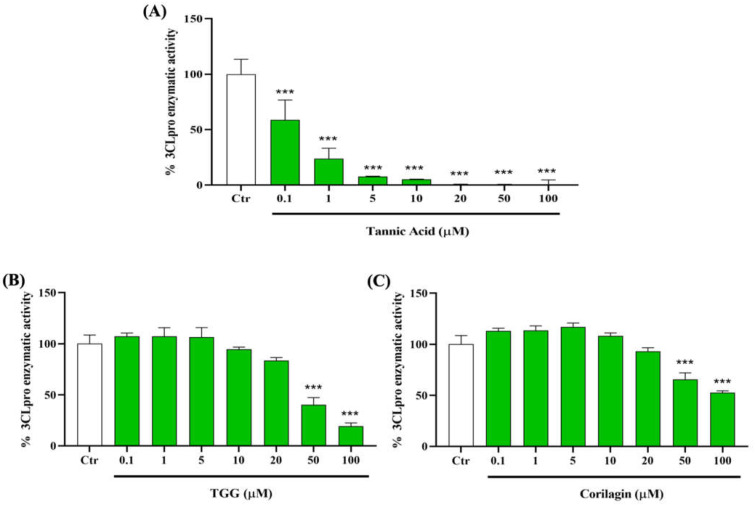
Inhibitory effects of TA, TGG, and corilagin on SARS-CoV-2 3-chymotrypsin like protease (3CLpro) activity. Different concentrations (0.1 to 100 µM) of (**A**) TA, (**B**) TGG, and (**C**) corilagin are tested on the activity of 3CLpro. The fluorescence units in control conditions are considered as 100%. Blank values are subtracted from all the readings before the conversion into percentage of activity. Results are expressed as mean ± SD (n = 3). Statistical analysis is performed using one-way ANOVA followed by the Tukey post hoc test with *** *p* < 0.001 compared to positive control wells.

**Figure 6 ijms-23-02643-f006:**
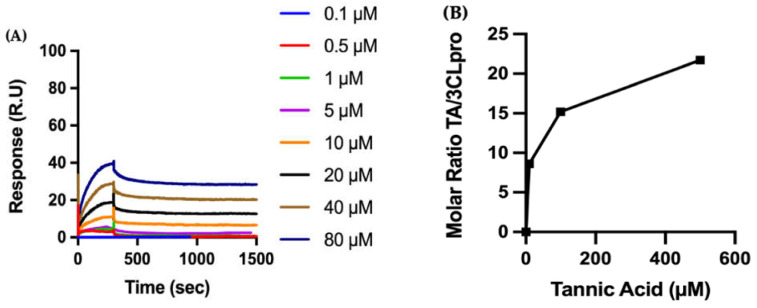
Biophysical characterization of the molecular interactions between the polyphenol TA on immobilized 3CLpro: (**A**) The recombinant protein 3CLpro is immobilized on a CM5 sensor chip, and increasing concentrations of TA are injected to evaluate binding kinetics by SPR. (**B**) 3CLpro is adsorbed to a gold QCMD sensor, and various concentrations of TA are flowed over the surface for 30 min. TA adsorption is expressed by the dimensionless molar ratio of adsorbed TA to adsorbed 3CLpro.

**Figure 7 ijms-23-02643-f007:**
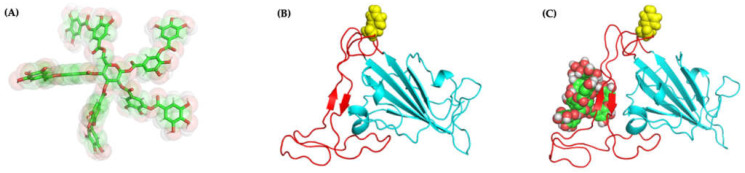
Molecular structures and docking of TA/RBD complex: one possible structure of (**A**) TA, (**B**) RBD (N501Y) (molecular dynamics (MD) 100 ns), and (**C**) TA/RBD (N501Y) complex (pose 1; highest docking binding affinity of −6.8 kcal/mol). The N501Y mutation is yellow.

**Figure 8 ijms-23-02643-f008:**
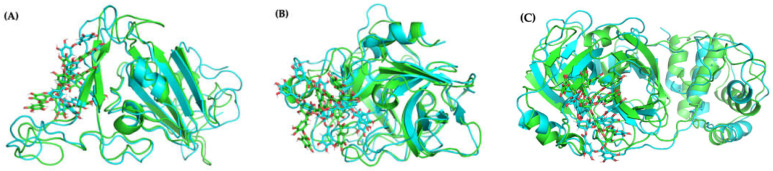
Molecular structures (pose 1) of: (**A**) TA/RBD, (**B**) TA/TMPRSS2, and (**C**) TA/3CLpro complexes, before (green) and after (turquoise) 1000-ns MD simulations.

**Figure 9 ijms-23-02643-f009:**
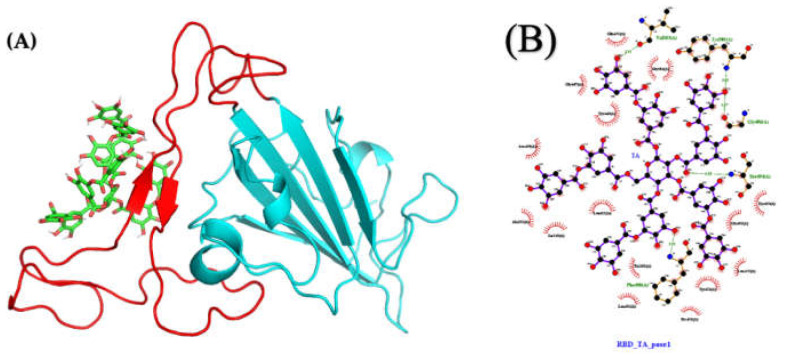
Molecular structures after 1000 ns of MD: (**A**) TA/RBD complex (pose 1; molecular mechanics Poisson–Boltzmann surface area (MMPBSA) binding free energy of −66 kcal/mol) and (**B**) ligand interaction map. The interaction map of TA with RBD (N501Y) is shown from the center of the biggest cluster computed on the convergence interval using the protein backbone atoms and ligand non-hydrogen atoms. The other contacts, defined by a distance smaller than 0.40 nm between the ligand and the protein, are shown as red arcs. H-bonds and their donor/acceptor distances are shown in green. The interaction map is generated using LigPlot [50,51].

**Figure 10 ijms-23-02643-f010:**
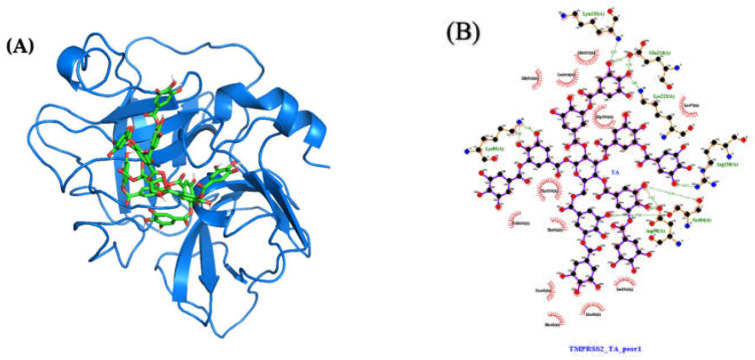
Molecular structures after 1000 ns of MD: (**A**) TA/TMPRSS2 complex (pose 1; MMPBSA binding free energy of −68 kcal/mol) and (**B**) ligand interaction map. The interaction map of TA with TMPRSS2 is shown from the center of the biggest cluster computed on the convergence interval using the protein backbone atoms and ligand non-hydrogen atoms. The other contacts, defined by a distance smaller than 0.40 nm between the ligand and the protein, are shown as red arcs. H-bonds and their donor/acceptor distances are shown in green. The interaction map is generated using LigPlot [50,51].

**Figure 11 ijms-23-02643-f011:**
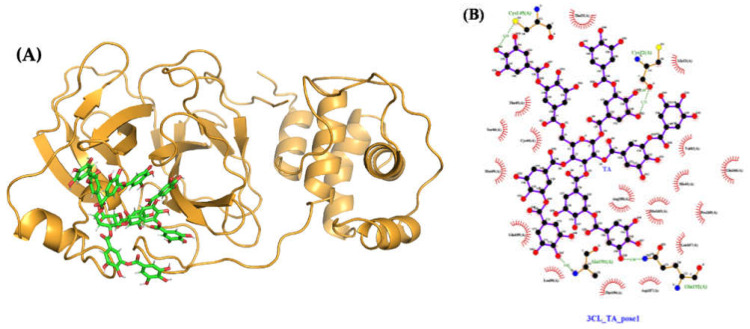
Molecular structures after 1000 ns of MD: (**A**) TA/3CLpro complex (pose 1, MMPBSA binding free energy of −65 kcal/mol); (**B**) ligand interaction map. The contact map of TA with 3CLpro is shown from the center of the biggest cluster computed on the convergence interval using the protein backbone atoms and ligand non-hydrogen atoms. The contacts, defined by a distance smaller than 0.40 nm between the ligand and the protein, are shown as red arcs. The interaction map is generated using LigPlot [50,51].

**Table 1 ijms-23-02643-t001:** Binding free energy between proteins (RBD, TMPRSS2, 3CLpro) and TA for the best poses found during docking. The MD MMPBSA binding free energy is computed over the interval 750 to 1000 ns using the g\_mmpbsa tools [49].

Binding Free Energy of TA/Protein Complex (kcal/mol)	
Protein (Pose)	MD (MMPBSA)
RBD (N501Y) (1)	−66
RBD (N501Y) (2)	−44
RBD (N501Y) (3)	−41
RBD (N501Y) (4)	−70
TMPRSS2 (1)	−68
TMPRSS2 (2)	−33
TMPRSS2 (3)	−57
TMPRSS2 (4)	−71
3CLpro	−65

## Data Availability

Not applicable.

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
