# Peer review of "Molecular Interactions of Tannic Acid with Proteins Associated with SARS-CoV-2 Infectivity"

_ijms, 2022, doi:10.3390/ijms23052643_

Round 1

Reviewer 1 Report

Please take care about abbreviations and explain them when they are first used (e.g. TGG in the abstract, or RBD alone as receptor binding domain at other places could be misleading). Figure/table captions should be self explanatory and sufficiently detailed, even in the supplementary material. For example, Figure S1, vertical axis explanation probably is: Results are expressed as Optical Density (OD450) measurements using a microplate reader with a 450nm filter. Also please specify your reader in experimental part. The starting protein pdb structures are not clearly defined in the 4.5.2. MD simulations part, for S-RBD perhaps 6M0J as in your previous work ? 6LU7 is given in parenthesis for 3CLpro, would be better (pdb code: 6LU7). I suggest to provide the sequences of all recombinant proteins used in this study, at least in the supplementary material, and mention if the same sequences were used in the in-silico work.

Author Response

We would like to thank the reviewers for their constructive comments and thoughts on the manuscript. We are confident that the new version of the article, as well as the discussion in the following document, will answer all questions and suggestions satisfactorily.

See attached answers to your questions/suggestions.

Kindest Regards

Roger

Reviewer 2 Report

The authors aimed to investigate experimentally the interactions between natural polyphenolic ligands and particularly tannic acid (TA), 1,3,6-tri-O-galloy-beta-D-glucose (TGG), and corilagin with proteins involved in relevant steps for the cellular entry (RBD (N501Y); the most frequent variant at the start of the study), transmembrane protease serine 2 (TMPRSS2) and replication (3CLpro) of the virus. For these reasons, they used a combination of experimental methods (biochemical ELISA assay, enzymatic assay, surface plasmon resonance (SPR), quartz crystal microbalance with dissipation monitoring (QCMD)) and numerical tools (molecular dockings, molecular dynamics and MMPBSA free energy calculations).

The study covers some issues that have been overlooked in other similar topics. The structure of the manuscript appears adequate and well divided in the sections. 

Moreover, the study is easy to follow, but few issues should be improved.

1-) The manuscript needs grammar correction. Please also check typos ad acronyms thorough the text, abstract included.

2-) Limitations of the study needs to better addressed.

3-) Conclusion Section: This paragraph required a general revision to eliminate redundant sentences and to add some "take-home message".

Author Response

(The authors gave the same response as above.)

Reviewer 3 Report

Dear Author,

This a very interesting scientific manuscript and it addresses a topic of high importance!

Please add tables for each section in order to reduce the length of sections.

KEYWORDS: SARS-CoV-2; COVID-19; Molecular dynamics; Polyphenols; Tannic acid; RBD; TMPRSS2; 3Clpro. The title words should not be repeated in Keywords.

ABSTRACT: (Please change): 3-chymotrypsin like protease (3CLPro) inhibitors

INTRODUCTION:

Line 40-42; SARS-CoV-2, is (please delete: is) a zoonotic corona virus of the SARS (severe acute respiratory syndrome), originated in China and resulted in a worldwide pandemic due to COVID-19 with more than 345 million known cases and 5.7 million deaths as of January 2022.

Line 42-46: New SARS-CoV-2 variants carrying out mutations are emerging with increasing infectivity that  facilitates their spread [1], e.g., the (insert the) SARS-CoV-2 B.1.617.2 (Delta) variant shows higher replication rate and transmissibility over B.1.1.7 (Alpha) [2], as well as B.1.1.529 (Omicron) with 30 mutations [3].

Line 49-52: In addition to the Spike-S protein receptor ACE2, the transmembrane protease serine 2 (TMPRSS2) and the intracellular replication/main protease (Mpro) (also called 3C-like protease-3CLpro) 3-chymotrypsin like protease (3CLPro) are required respectively, for its cell entry and replication.

Line 79-81: Since the early COVID-19 pandemic, most studies have (insert have) focused on the inhibition of the RBD/ACE2 binding [29], but natural polyphenols could also target two proteolytic enzymes : the transmembrane TMPRSS2 and the intracellular 3CLpro.

DISCUSSION

Line 366-368: The mutagenic ribonucleoside Molnupiravir [54] and the 3CLpro protease inhibitor recently became FDA-approved.

Line366-368: The (insert: The) projected cost is around 500$ (Paxlovid) and $700 (Molnupiravir) per person for a 5-day course. It is thus  important to identify a multitargeted and low-cost drug for the management of SARS-CoV-2–induced infection without or with limited side effects.

Line 451-453: Recent A recent review summarizes numerous results on SPR biosensing of SARS-CoV-2 [64,65]. In this work, both SPR and QCMD methods show high  affinity between TA and the human ACE2 receptor, the TMPRSS2 and 3CLpro.

Line 457-462: However, Pitsillou et al.[32]  showed MMPBSA binding free energy ofcyanidin-3-O-glucoside/3CLpro of -50.8 and –  42.1 kcal/mol with protomer A and B, respectively, whereas Singh et al.[66] showed highest free energy among three phenolic ligands (mangiferin, glucogallin and phlorizin) with 3CLpro (-9.65 ± 3.33 kcal/mol), where  all these ligands´molecular weights the moleculr weights of all these ligands (MW) (332.26- 449.38 g/mol) are four to five times smaller than that of TA (1701.18 g/mol).

Please, highlight the strengths and weaknesses (or limitations) of your methods beyond “the lack in mimicking SARS-Cov-2 -induced cellular or animal models in complex microenvironments”.

REFERNCES

Please see the reference:

Jurica Novak  Vladimir A Potemkin  A new glimpse on the active site of SARS-CoV-2 3CLpro, coupled with drug repurposing study. Mol Divers, 2022 Jan 10;1-15.doi: 10.1007/s11030-021-10355-8.

Author Response

(The authors gave the same response as above.)
